# Rice Yield Prediction in Hubei Province Based on Deep Learning and the Effect of Spatial Heterogeneity

Shitong Zhou [1], Lei Xu [1] and Nengcheng Chen [1,2,*]

1   National Engineering Research Center for Geographic Information System, School of Geography and Information Engineering, China University of Geosciences (Wuhan), Wuhan 430074, China
2   Hubei Luojia Laboratory & State Key Laboratory of Information Engineering in Surveying, Mapping, and Remote Sensing, Wuhan University, Wuhan 430079, China
*   Correspondence: chennengcheng@cug.edu.cn; Tel.: +86-138-8601-9231

**Abstract:** Timely and accurate crop yield information can ensure regional food security. In the field of predicting crop yields, deep learning techniques such as long short-term memory (LSTM) and convolutional neural networks (CNN) are frequently employed. Many studies have shown that the predictions of models combining the two are better than those of single models. Crop growth can be reflected by the vegetation index calculated using data from remote sensing. However, the use of pure remote sensing data alone ignores the spatial heterogeneity of different regions. In this paper, we tested a total of three models, CNN-LSTM, CNN and convolutional LSTM (ConvLSTM), for predicting the annual rice yield at the county level in Hubei Province, China. The model was trained by ERA5 temperature (AT) data, MODIS remote sensing data including the Enhanced Vegetation Index (EVI), Gross Primary Productivity (GPP) and Soil-Adapted Vegetation Index (SAVI), and a dummy variable representing spatial heterogeneity; rice yield data from 2000–2019 were employed as labels. Data download and processing were based on Google Earth Engine (GEE). The downloaded remote sensing images were processed into normalized histograms for the training and prediction of deep learning models. According to the experimental findings, the model that included a dummy variable to represent spatial heterogeneity had a stronger predictive ability than the model trained using just remote sensing data. The prediction performance of the CNN-LSTM model outperformed the CNN or ConvLSTM model.

**Keywords:** rice; crop yield prediction; CNN-LSTM; spatial heterogeneity; Google Earth Engine; deep learning

## 1. Introduction

Crop yield prediction is an essential task for the decision makers at national and regional levels (e.g., the EU level) for rapid decision making [1]. Farmers' ability to plan and be ready for planting can be improved with timely and accurate crop production information. It is of great importance to ensure agricultural food security.

Remote sensing has been extensively employed in agricultural applications, such as crop cover categorization, drought stress estimate and yield prediction, because remote sensing data are generally open source and reasonably priced [2]. The vegetation index, calculated based on remote sensing data, reflects the state of chlorophyll within the crop and correlates with the crop growth status. The Normalized Vegetation Index (NDVI) is considered to be the primary index exercised to monitor crop conditions [3]. Other indices, the Enhanced Vegetation Index (EVI) [4], the Green Normalized Vegetation Index (GNDVI) [5], the Green Leaf Area Index (GLAI) [6], Gross Primary Productivity (GPP) [7], the Normalized Differential Water Index (NDWI) [8] and the Soil-Adjusted Vegetation Index (SAVI) [9], have also been used for crop yield prediction. The satellite remote sensing data commonly used to study the calculation of vegetation indices are mainly MODIS

data, Landsat and Sentinel-2 satellite data [10]. Soil and climate-related variables are also commonly used for yield forecasting, such as land cover [11], an integrated drought index [12], soil pH, soil moisture [13], precipitation, air temperature and humidity [14], etc.

Based on remote sensing data, there are two main approaches to crop yield prediction: crop process simulation models and empirical statistical models [15]. The agricultural process simulation model can effectively replicate the physical processes of plant growth based on physiological knowledge of plant growth. It is the basic crop yield prediction model. Due to the inclusion of growth and environmental elements, these models are challenging to employ for large-scale forecasts. Empirical statistical models do not directly consider the physiological mechanisms of the plant. It is relatively simple and requires a smaller set of parameters and input variables. Empirical statistical models are more widely used in the field of crop yield prediction. Machine learning and deep learning algorithms can improve the accuracy of crop yield prediction compared to traditional linear regression methods. These more sophisticated and intelligent algorithms extract information automatically. Algorithms, such as support vector machines (SVM) [16], decision tree regression (DTR), and convolutional neural networks (CNN) [17], have been successfully applied.

Xu presented a comparative review of statistical, physical and artificial intelligence methods for spatiotemporal forecasting problems. Statistical methods are not conducive to advanced feature extraction and long-term memory modeling. The model structure and parameterization of physical models are imperfect and computationally intensive. Artificial intelligence models require elaborate training but have advantages for complex nonlinear and non-Gaussian problems [18]. For linear regression methods, machine learning methods and deep learning methods, Cao et al. selected EVI and SIF (solar-induced chlorophyll fluorescence) as feature factors for comparison experiments to predict rice yield in China [19]. The results showed that the machine learning (RF) and deep learning (LSTM) methods performed significantly better than the linear regression. Compared with traditional linear regression methods, machine learning methods and deep learning methods can better extract the nonlinear relationship between input variables and rice yield. In addition, the result showed that the deep learning method outperformed the machine learning method in rice yield estimation, in part because the recurrent neural network structure in LSTM can fuse the nonlinear relationship between rice yield and environmental factors. Nabila used five regression models to forecast wheat yields in two provinces of Algeria [20]. It was found that ANN, RF and ELM based on machine learning methods outperformed DNN regression models in terms of prediction performance when the input variables were fewer, and when the number of input variables was more, the DNN model performed better than the other models. This is because deep learning methods typically outperform ordinary machine learning models for datasets with larger amounts of data. Juan et al. also compared traditional machine learning methods (RF) with deep learning methods (DNN, 1D-CNN and LSTM) based on winter wheat production areas in China [21]. The results showed that machine learning methods were not always worse than deep learning methods at the county and regional scales. To some extent, the RF model outperforms other deep learning models. He speculated that one of the reasons might be that the deep learning method requires a large dataset for training, and that the feature variables selected for the experiments are strongly correlated with wheat yields and cannot fully exploit the advantages of the deep learning method for automatic feature extraction. The choice of model approach depends on the problem, the data and the corresponding requirements [22]. Both machine learning and deep learning have been widely used in the field of crop yield prediction. Machine learning and deep learning have a powerful ability to fit nonlinear relationships [23]. These two methods have their own advantages. For significantly large datasets, deep learning methods are better able to take advantage of handling complex relationships and automatically extracting features, and for ordinary datasets, the prediction performance of traditional machine learning methods is no worse than that of deep learning methods. Xiang conducted comparative experiments based

on CNN-LSTM hybrid models for NDVI, EVI, LST and soil moisture and found that EVI performed the best. The results also showed that the hybrid model had less prediction error than a single CNN or LSTM [24]. Ji et al. investigated the ability of combining physical indicators with NDVI remote sensing parameters to predict yield. The results showed that the multivariate regression model using the phenological indicators and NDVI was better than the NDVI univariate regression model in predicting yield [25]. Nevavuori et al. proposed three model architectures: CNN-LSTM, ConvLSTM and 3D-CNN using drone sequences and weather data. The results showed that 3D-CNN performed the best among all model architectures [26]. Wang et al. proposed a CNN and LSTM dual branch model: an LSTM network branch based on remote sensing and meteorological data; a static soil feature model branch constructed using CNN. The evaluation results performed well, and the results also showed that yield prediction could be achieved at least one month before harvest [27]. Fernandez et al. used 3D-CNN to train the feature data based on Sentinel-2 satellite data, meteorological data and soil data. The results showed that the yield prediction performance of 3D-CNN was better than 2D-CNN [28]. Yang et al. used convolutional neural networks (CNNs) to learn the feature factors associated with rice growth. The results showed that the neural network trained by RGB and multispectral datasets performed better than the vegetation-index-based regression model in yield estimation [29]. Yaramasu et al. proposed a model architecture based on a bidirectional ConvLSTM network. Spatial features were first extracted using a convolutional neural network, and then the extracted spatial features were analyzed in time using a recurrent neural network (RNN) based on LSTM. The early crop map was predicted based on satellite data from Nebraska [30]. Sun et al. converted MODIS surface reflectance and surface temperature into histograms based on Google Earth Engine (GEE). Combined with weather data, CNN-LSTM was used as the model architecture to predict the annual soybean yield in the United States [31]. The remote sensing data were downloaded based on the GEE platform by Shelestov et al. Multiple classifiers in GEE were compared to generate high-resolution crop classification maps over large areas [32]. Han et al. developed a modeling framework based on GEE to integrate remote sensing data, meteorological data and soil data. Eight typical machine learning models were tested [33].

With the continuous development of remote sensing technology, the resolution of satellite remote sensing images has been increasing, and more and more accurate information about crops can be observed. Crop yield prediction at large spatial and temporal scales has a great demand for downloading and preprocessing satellite remote sensing data with multiple time series. The emergence of deep learning, big data and cloud computing technologies has solved the problems of tedious image downloading, storage and integration, and time-consuming data computing in the traditional remote sensing image processing. Google Earth Engine (GEE) is provided by Google, a cloud-based geospatial processing platform for analyzing planetary-scale geospatial data [34]. The platform has access to most satellite data and provides tools to process and analyze the data online. Based on the GEE platform, vegetation index data such as NDVI and EVI can be downloaded directly. It is also possible to perform wave synthesis calculations, normalization and other operations.

Generally, when selecting crop-growth-related feature variables, only the correlation between regional yield and each feature is considered. However, there are differences in land use, soil type and cover within the region. Such differences are the heterogeneity and complexity in spatial distribution within the region, i.e., spatial heterogeneity. Spatial heterogeneity is an important research theory in the field of ecology, and has gradually been applied to the field of human–economic geography [35]. Spatial heterogeneity can, to some extent, affect the correlation between crop yield and various types of characteristic data, which in turn affects the final prediction. Currently, the effect of spatial heterogeneity has been less considered in the existing studies in the field of crop yield.

Through the previous scholars' studies, we can learn that deep learning models have greater advantages over traditional linear regression methods in dealing with complex nonlinear relationships. Moreover, deep learning models perform better than machine

learning models when dealing with a larger number of datasets. Among the many modeling approaches in deep learning, CNN and LSTM are widely used in the field of crop yield prediction. Many studies have shown that the CNN-LSTM hybrid model approach has higher accuracy in solving crop yield prediction problems. For the county-level rice data in Hubei Province, China, which has a long time interval and a large amount of data, we chose the deep learning model as the prediction model for prediction. To evaluate the accuracy results of the hybrid CNN-LSTM model on the rice yield prediction problem and to verify its prediction performance, experiments were conducted. In this paper, the prediction ability of the CNN-LSTM hybrid model was tested based on county-level rice yield in Hubei Province, China, and the CNN model and ConvLSTM model were used as comparison experiments. The GEE platform was utilized to download satellite remote sensing information. The images were synthesized and processed to convert them into normalized histograms. As input data for training the model, vegetation indices, climatic indicators and custom parameters indicating spatial heterogeneity were used. Rice production data of Hubei Province from 2000–2019 were used for validation.

The main objectives of this study are: (1) Exploring a satisfactory method for county-level rice yield forecasting. To verify the prediction ability of the CNN-LSTM model, the CNN model and the ConvLSTM model are used for comparison. (2) Discuss the effect of spatial heterogeneity on the results of rice yield prediction. We add a custom dummy variable and evaluate whether it can improve the final prediction accuracy.

## 2. Materials and Methods

### 2.1. Study Area

The study area chosen for this paper is the rice planting area in Hubei Province, China. The total planted area is about 2.28 million hectares and the total production is about 18.64 million tons. It is one of the most important main rice-producing areas in the country. The topography of Hubei Province varies greatly from east to west, and the natural conditions are complex. The western region has higher topography and lower average annual temperature, so less area is planted for rice. The central and eastern regions are dominated by double-season rice. Figure 1 shows the rice planting area in Hubei Province. In this study, county-level rice planting areas in 14 cities in Hubei Province were selected.

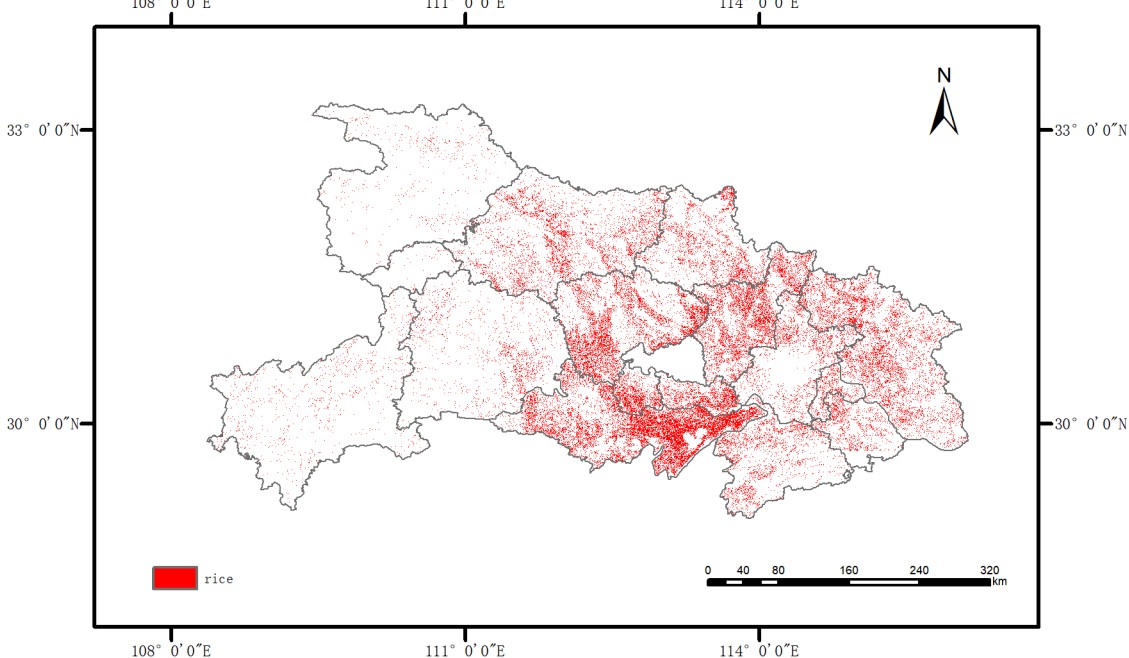

**Figure 1.** Rice planting area in Hubei Province.

## 2.2. Data

MODIS EVI data, GPP data, SAVI data, air temperature data and a dummy variable representing spatial heterogeneity were selected as input variables for the study. Rice yield data for Hubei Province from 2000 to 2019 were used for validation. Rice mask layer and Hubei county boundary data were used as auxiliary data. The satellite remote sensing data were collected from April to January, the time when double-season rice is planted and harvested. The following is a description of the relevant data.

### 2.2.1. Hubei Province Yield Data

County-level rice yield data for Hubei Province from 2000 to 2019 were obtained from the Hubei Provincial Bureau of Statistics and the China Economic and Social Data Research Platform in tons. The yield data were used as training and validation of the model.

### 2.2.2. Rice Mask Layer

The source of the rice mask layer data was the research data of Mr. Shen Yonglin, School of Geography and Information Engineering, China University of Geosciences (Wuhan). Mr. Shen classified the major crop types in Hubei Province into rice, corn, cotton, wheat and rapeseed [36]. In this study, the rice planting area was selected as the mask layer. The main purpose of the mask data is to exclude the interference of other non-rice vegetation when calculating the vegetation index. The spatial resolution of this dataset is 30 m. The overall classification accuracy is around 85%, which achieves a good classification result.

### 2.2.3. County Boundary Data

Hubei county boundary data were downloaded from AliCloud's map selection site. To receive the data in the right file format, we used a free online conversion tool.

### 2.2.4. MODIS Data

The MODIS sensor is carried by two satellites, Terra and Aqua. Data products are timely and abundant. In the field of crop yield prediction, MODIS data are used more commonly. In this study, *EVI*, *GPP* and *SAVI* from MODIS data were selected.

Enhanced Vegetation Index *EVI*, an improved and optimized version of NDVI, can more accurately reflect vegetation growth changes in areas with high vegetation cover [37]. MOD13A2 V6 product provides *EVI* with an average of 16 days and 1 km resolution. Equation (1) shows the formula for *EVI*, where *NIR*, *RED* and *BLUE* are the near-infrared and red light bands provided by the MOD13A2 V6 product.

$$EVI = 2.5 \times \frac{NIR - RED}{NIR + 6.0RED - 7.5BLUE + 1} \tag{1}$$

Gross Primary Productivity *GPP*, which represents the total amount of organic carbon fixed by photosynthesis per unit time per unit area of green plants, is widely used in crop yield estimation [38]. The MOD17A2H V6 product provides 8 days of cumulative total primary productivity data with 500 m resolution. Equation (2) shows the formula for *GPP*, where *NIR*, *RED* and *BLUE* are the near-infrared and red light bands provided by the MOD13A2 V6 product. Equation (2) shows the formula for GPP, where *APAR* is photosynthetically active radiation, *FPAR* is the photosynthetically active radiation absorption ratio by vegetation, and $\varepsilon$ is the realistic light energy utilization based on the *GPP* concept.

$$GPP = APAR \times FPAR \times \varepsilon \tag{2}$$

The soil-adjusted vegetation index *SAVI* is also commonly used for crop yield prediction with an increased soil adjustment factor to mitigate the effect of soil noise on the vegetation index [39]. MODIS products do not provide the relevant data directly. It is obtained by waveband calculation. Equation (3) shows the formula for *SAVI*, where *L* is

the soil conditioning factor, and *NIR* and *RED* are the near-infrared and red light bands provided by the MOD13A2 V6 product, respectively.

$$SAVI = (1 + L)\frac{NIR - RED}{NIR - RED + L} \tag{3}$$

2.2.5. Weather Data

Air temperature data were provided by ERA5. ERA5 is the fifth generation of the global climate reanalysis dataset released by the European Centre for Medium-Range Weather Forecasts (ECMWF). The ERA5 dataset provides an accurate description of climate (e.g., temperature, precipitation and wind speed) from local to global scales in the past and has been applied in areas such as yield prediction and disaster monitoring [40]. Average air temperature data were downloaded with a spatial resolution of 2 m based on the GEE platform.

2.2.6. Spatial Heterogeneity Variables

Considering that spatial heterogeneity can affect the rice yield prediction results to some extent, in this study, we defined a dummy variable as a characteristic factor representing spatial heterogeneity to be input into the model. This dummy variable was specifically defined as the number of each county.

*2.3. Preprocessing in GEE*

As for deep-learning-based prediction processing, most methods prefer to select the mean or VI of regions as features because these methods have low computational complexity [31]. The minimum outsourcing rectangle size varies from county to county, and the large amount of raw remote sensing data fed directly into the deep learning model requires enormous computational power. Thus, we converted the remote sensing images into histograms by referring to others' research in order to improve the computational efficiency. In this study, remote sensing data were downloaded based on the GEE platform, and the raw images were preprocessed. Figure 2 shows the flow of data processing based on the GEE platform. The preprocessing steps are as follows.

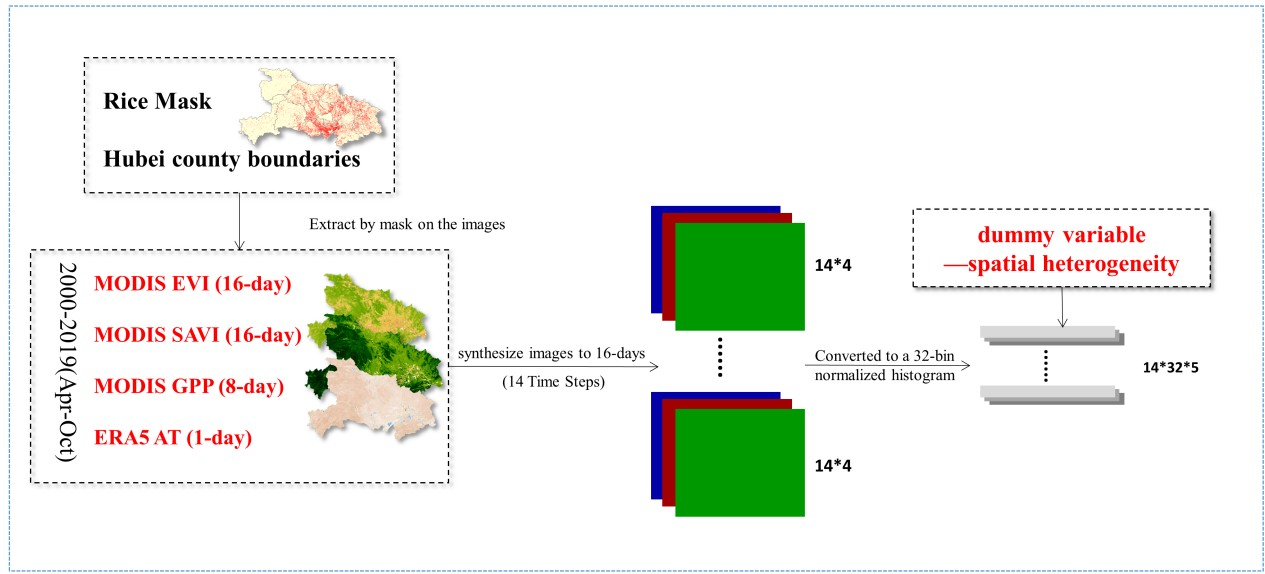

**Figure 2.** GEE processing flow.

1.  Download data. Download remote sensing data from 2000 to 2019, April to October. A 16-day synthesis of 8 days of MODIS GPP data and daily air temperature data from ERA5 was performed. Alignment with MODIS EVI and MODIS SAVI;

2. Rice masks and county boundary layers. The rice mask layer is used to process the remote sensing data and eliminate the interference from other vegetation on the ground. The county boundary data of Hubei province are imported into GEE to extract remote sensing data of each county;

3. Convert the histogram. GEE provides a convenient and fast API to convert image collections into county-level 32-bin normalized histograms;

4. A dummy variable. After numbering each county in Hubei Province, the output is a 32-bin histogram in a uniform format, which is added as a factor to the feature;

5. In this study, annual time steps of 14, with 5 bands per time step, were converted to histograms and then input into the model. The format of the input variables is $32 \times 5 \times 14$. The corresponding county-level yield was assigned to each input variable based on the obtained rice yield statistics in Hubei Province.

Model Architecture

1. CNN. Convolutional neural network (CNN) is a neural network with convolutional structure. The basic components are input layer, convolutional layer, pooling layer, fully connected layer and output layer. Convolutional layers are linked to the input layer using local weighting and weight sharing. The features of the input data are extracted by convolutional kernels. The pooling layer reduces the amount of data for convolution operations. After the convolution and pooling layers, one or more fully connected layers are usually connected. Fully connected layers can integrate local information with category differentiation in the convolutional or pooling layers [41]. The output value of the last fully connected layer is passed to the output layer. The main component of the CNN model is the convolutional operation. CNN uses a convolutional kernel applied to the input variables to produce a set of spatial features of the input data by convolutional operations. In this paper, we set up two convolutional layers, Conv2D. The first layer was set up with 32 filters and the convolutional kernel size was $3 \times 3$. The second layer was set up with 64 filters and the convolutional kernel size was $3 \times 3$. The pooling layer used the maximum pooling method.

2. ConvLSTM. The long short-term memory network (LSTM) is a modified version of the recurrent neural network (RNN). Unlike CNNs, the neurons in RNNs have a feedback structure. This feedback structure enables the previous data to receive the influence of the later data. Therefore, recurrent neural networks have better performance when dealing with temporally correlated sequential data. LSTM effectively improves the problem of gradient explosion, which exists in recurrent neural networks and makes it difficult to learn the relationship between long interval data, by filtering the information obtained through the gate function. Convolutional LSTM (ConvLSTM) [42] is a model that combines features of convolutional and sequential models into a single architecture. Use the convolutional layer as the gate function of the LSTM [26]. ConvLSTM uses the three-gate control structure of LSTM [43] and uses convolutional operations to extract spatial features. The ConvLSTM model used in this paper was set up with 3 ConvLSTM2D layers and a convolutional kernel size of $1 \times 2$.

3. CNN-LSTM. CNN can learn relevant features from images. The LSTM network performs well on data processing of long time sequences. The CNN-LSTM model used in this study consists mainly of a two-dimensional convolutional neural network and an LSTM network. The CNN first extracts spatial features and then passes the extracted spatial features to the LSTM network. The input to the model is based on the $32 \times 5 \times 14$ feature variables generated by GEE preprocessing. Figure 3 shows the CNN-LSTM model architecture we used.

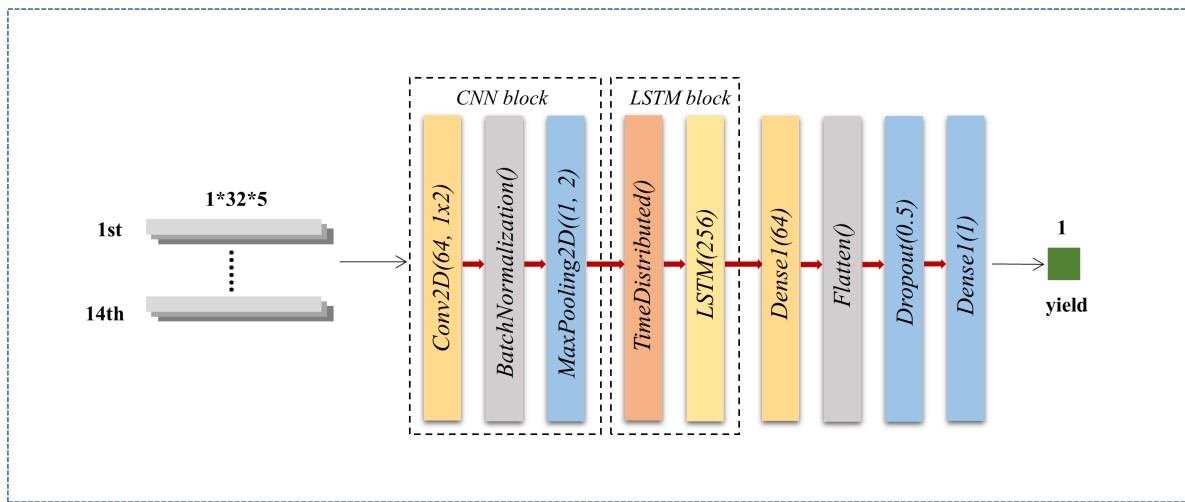

**Figure 3.** The architecture of the CNN-LSTM model.

The output value is the predicted rice yield value. The Conv2D layer of CNN is set with 64 filters and the convolutional kernel size is $1 \times 2$. The convolution layer connects the BatchNormalization layer. Then there is a MaxPooling2D layer with a kernel of $1 \times 2$. The batch normalization process allows the CNN to resist vanishing gradients during training, which can reduce training time and lead to better performance [44]. The Conv2D layer is applied to every temporal slice of the inputs for feature extraction via the TimeDistributed wrapper. Then the output is passed to the LSTM layer. The LSTM layer is set with 256 neurons. The Dense layer is set with 64 neurons. After that, the Flatten layer is connected and some of the neurons during training are randomly turned off with 0.5 dropout probability. The dropout strategy can prevent overdependence between neurons. Finally, the predicted yield values are output using a one neuron Dense layer. The activation functions of the model are chosen as linear rectifier function (Relu) and hyperbolic tangent function (Tanh). The optimization function of the learning rate uses Adam's algorithm to provide an adaptive learning rate.

*2.4. Evaluation*

In this study, CNN and ConvLSTM were used as comparison experiments of CNN-LSTM models. The division of the training set, validation set and test set was 14:3:3. Data from 2000 to 2013 were used as the training set for training, and data from 2014 to 2016 were used as the validation set to verify the output models. Based on the trained model, rice yields were predicted for 2017, 2018 and 2019. Comparing the predicted yield with the statistical yield data allows the performance of the prediction model to be evaluated year by year as well as the overall prediction effect over 3 years. Root mean square error *RMSE*, mean absolute error *MAE*, percent error *PE* and correlation coefficient *R* were chosen as evaluation metrics. The coefficient of determination R2 was also used in the study to evaluate the extent of spatial variation in predicted and observed yields. Equations (4)–(7), respectively, give the equations for *RMSE*, *MAE*, *PE* and *R*, where $y_i$ is the predicted value, $\hat{y}_i$ is the statistical value and *n* is the number of samples

$$RMSE = \sqrt{\frac{\sum_{i=1}^{n}(y_i - \hat{y}_i)^2}{n}} \tag{4}$$

$$MAE = \frac{\sum\limits_{i=1}^{n}|y_i - \hat{y}_i|}{n} \tag{5}$$

$$PE = \frac{|y_i - \hat{y}_i|}{\hat{y}_i} \cdot 100\% \tag{6}$$

$$R = \frac{\sum\limits_{i=1}^{n} (y_i - \overline{y}_i)(y_i - \overline{\hat{y}}_i)}{\sqrt{\sum\limits_{i=1}^{n} (y_i - \overline{y}_i)^2}\sqrt{\sum\limits_{i=1}^{n} (y_i - \overline{\hat{y}}_i)^2}} \tag{7}$$

## 3. Results

### 3.1. Comparison of the Three Models

Three sets of models were trained by adjusting the parameters, from which a set of models with the best performance was selected. Figure 4 shows the Loss and MAE of the model validation set. During the training process, the CNN-LSTM hybrid model performed better than the CNN model and the ConvLSTM model after the epoch reached 250 times. The training effect of the CNN model is best when the epoch is less than 150 times, and the training effect of the CNN-LSTM model and ConvLSTM model is not much different. When the epoch is in the interval of 150 to 250 times, the CNN model is comparable to the CNN-LSTM model training effect.

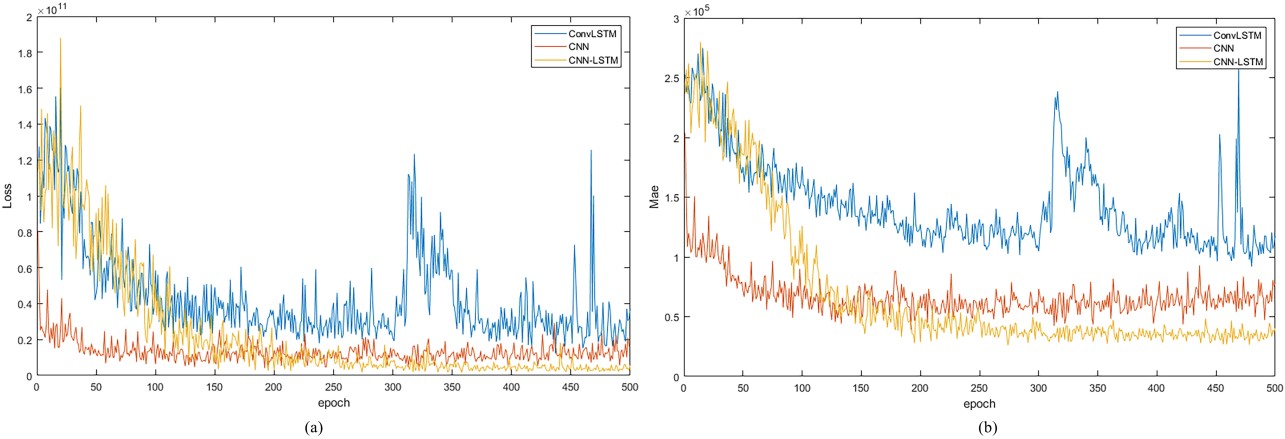

**Figure 4.** Loss and MAE of the three model validation sets. (**a**) is the validation set Loss, (**b**) is the validation set MAE.

Table 1 shows the rice yield prediction performance of different models (CNN, ConvLSTM and CNN-LSTM). RMSE, MAE and correlation coefficient R were used as evaluation indicators between predicted and observed yields. The result shows that the best performing model architecture among the three models is the CNN-LSTM hybrid model. Compared with other trained models, the CNN-LSTM hybrid model has obvious advantages in rice yield prediction. Compared with the CNN model and the ConvLSTM model, the RMSE of the CNN-LSTM model was reduced by 20.4% and 46.5%, the MAE was reduced by 19.9% and 54.5%, and the correlation coefficient R was improved by 9.88% and 29.5%, respectively. The worst-performing model among the three models is the ConvLSTM model. Its performance is reflected in the evaluation metrics, which is clearly inferior to the other two models.

**Table 1.** Model performance of rice yield prediction measured by root mean square error RMSE, mean absolute error MAE and correlation coefficient R measures.

| Model | Test RMSE (t) | Test MAE (t) | Test R - |
|---|---|---|---|
| CNN | 112,877 | 65,943 | 0.850 |
| ConvLSTM | 168,056 | 115,973 | 0.721 |
| **CNN-LSTM** | **89,878** | **52,802** | **0.934** |

In response to the study's objective (1), we conclude that the CNN-LSTM hybrid model is a more satisfactory model for rice yield prediction compared with the CNN and ConvLSTM models. The CNN-LSTM hybrid model exhibits better prediction performance in all three evaluation metrics.

Figure 5 shows a comparison of the yield distribution map between the statistical data of rice production in Hubei Province and the predicted production at the county level.

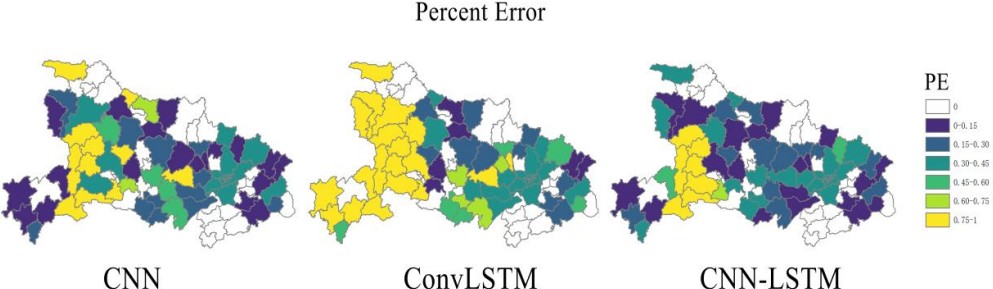

**Figure 5.** Percentage error map of the forecasts for 2019 for the three models.

*3.2. Accuracy of CNN-LSTM Hybrid Model*

The percentage error PE was used as a comparison indicator. The first row is the rice production data from 2017 to 2018 in Hubei Province, the second row is the predicted county-level rice production based on the CNN-LSTM hybrid model, and the last row is the county-level error percentage calculated based on the predicted and observed yields. The dark color means low production and percentage, and vice versa.

In addition to RMSE and PE, we used the coefficient of determination $R^2$ to evaluate the predicted and observed yields, as shown in the scatter plot in Figure 6.

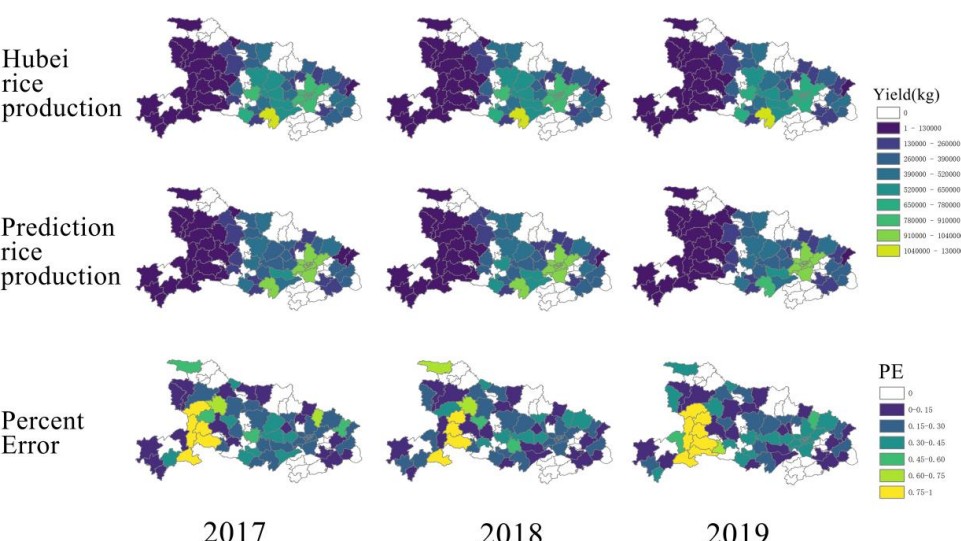

**Figure 6.** Map of rice production in Hubei Province, projected production and percentage error (PE) from 2017 to 2018.

According to Figure 5, it can be seen that there is high consistency between the predicted production and the statistical data on rice production in Hubei Province. Rice production in Hubei Province is mainly concentrated in the central and eastern parts of the country. Rice production is lower in the northwestern mountainous areas. Most predictions have an error percentage of less than 30%. However, some extremely high forecast errors occur mainly in the northwestern mountainous areas, such as the Shennongjia Forest area, and parts of Yichang and Enshi, which appear bright yellow. The occurrence of these higher prediction errors may be related to the topographic relief and local climatic environment.

Rice growth is influenced by the natural climate. Temperature and precipitation, among others, directly affect the growth and distribution of rice [45]. The spatial and temporal distribution of precipitation in Hubei Province is uneven, with a nearly three-fold difference between the north and south of the province. The Jianghan basin is rich in water. However, the northern part of Hubei Province is more drought-prone. There were consecutive droughts in the winter of 2010 and the spring of 2011, which are rare in history [46]. The Shennongjia forest area is at a higher elevation than the central-eastern Jianghan Plain area, and the climate is very different from that of the plain areas. The area under rice cultivation is relatively small and the rice yield is also at a low level in the province. There are also differences in rice irrigation efficiency in different regions of Hubei Province [46]. In addition to the amount of precipitation that affects rice irrigation efficiency, there are also effects of education level, sown area and population size in different regions on rice irrigation efficiency. Rice irrigation in the Jianghan Plain is more efficient, and the rice cultivation area is more extensive. However, in the northwest and southwest of Hubei Province, rice irrigation is less efficient and rice cultivation is sparse, resulting in lower rice yields. Rice needs to be in the right temperature range to grow properly. If rice is exposed to a high temperature above 35 °C during the growing period, it will be prone to heat damage [47]. Heat damage in the Jianghan Plain is low and short-lived, and most of the heat damage is mild. There is more heat damage in the Three Gorges area and western Hubei Province and most of this heat damage is severe. These reasons contribute to yield differences and forecast errors between the central-eastern region and the northwestern mountainous region.

From 2017 to 2019, the R2 shows that the forecasted yield can explain 86%, 88% and 87% of the variance in the observed yield. It is demonstrated that the CNN-LSTM hybrid model can indeed effectively predict county-level rice yield.

You can see in Figure 7. scatter plots of predicted and observed yield from 2017 to 2019.

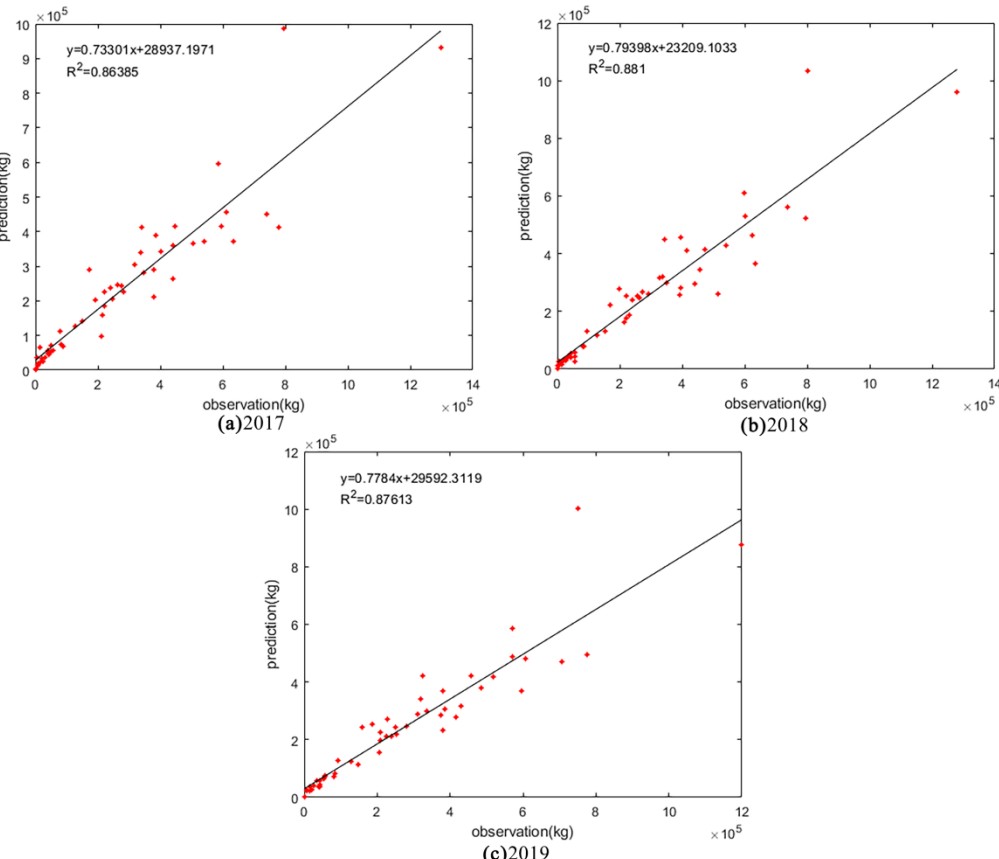

**Figure 7.** Scatter plots of predicted and observed yield from 2017 to 2019.

### 3.3. Impact of the Dummy Variable on Prediction Accuracy

In this study, we considered the effect of spatial heterogeneity within regions on county-level yield forecasting by adding a custom dummy variable to the input variables. The dummy variable is the number of each county in Hubei Province. The root mean square error RMSE, mean absolute error MAE and correlation coefficient R were used as evaluation indicators to assess the performance of the dummy variable in the rice yield prediction experiment.

Figure 8 shows the evaluation results of the three groups of model comparison experiments before and after the addition of the dummy variable.

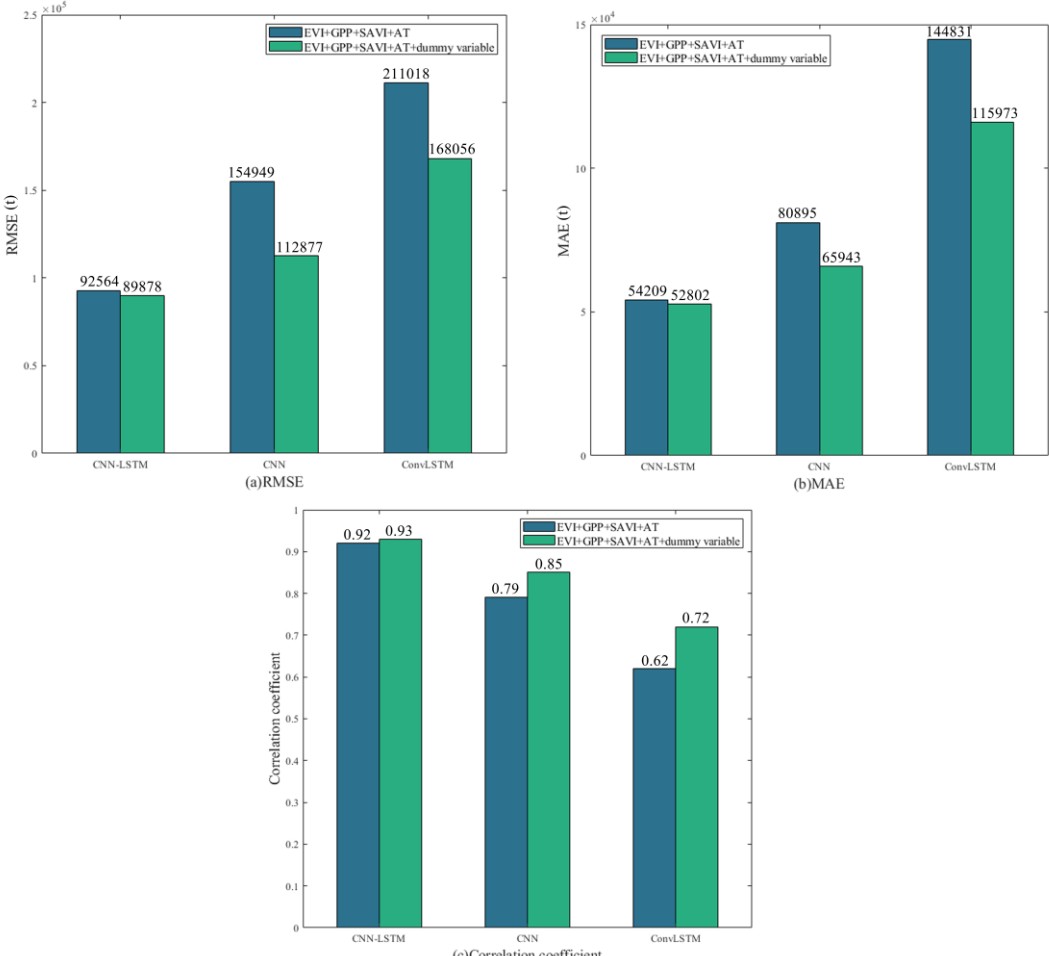

**Figure 8.** Comparative experimental results of the three groups of models regarding the dummy variable. (**a**) is the comparative result of RMSE after adding a dummy variable. (**b**) is the comparison result of MAE after adding a dummy variable. (**c**) is the comparison result of the correlation coefficient after adding a dummy variable.

As can be seen from the figure, the prediction accuracy of the CNN model, the ConvLSTM model and the CNN-LSTM hybrid model are all improved after the addition of the dummy variable. The CNN-LSTM hybrid model performed best in prediction after adding the dummy variable. The CNN model and the ConvLSTM model showed the most significant accuracy improvement after adding the dummy variable. RMSE decreased by 27.5% and 20.4%, MAE decreased by 18.4% and 20.0%, and the correlation coefficient R improved by 7.6% and 16.1%, respectively.

In response to the study's objective (2), the experimental results showed that spatial heterogeneity has an effect on rice yield prediction, and the dummy variable was signif-

icantly helpful to improve the accuracy of prediction results. It is important to consider spatial heterogeneity in yield prediction experiments.

## 4. Discussion

In this study, we evaluated the predictive performance of the deep learning model architecture on county-level rice yields. Remote sensing data and climate data of the rice growing period from 2000 to 2019 in Hubei Province, China, were used. On this basis, the effect of spatial heterogeneity was taken into account by adding the dummy variable of county numbering together as an input variable for prediction. We converted the remote sensing images into normalized 32-bin histograms based on GEE, which can facilitate the input of deep learning models. We designed and trained three different deep learning model architectures: CNN, ConvLSTM and CNN-LSTM. We obtained the model architecture with the best prediction performance by tuning the parameters. Data from 2000 to 2016 were used for training and validation, and data from 2017 to 2019 were used to test the predictive power of the model.

Among these model architectures, the CNN-LSTM hybrid model has the best prediction performance. The prediction effects of the three model architectures are given in Section 3.1. The CNN-LSTM hybrid model has the best performance in the three evaluation metrics of root mean square error RMSE, mean absolute error MAE and correlation coefficient R. Compared with the CNN model and the ConvLSTM model, the RMSE of the CNN-LSTM model was reduced by 20.4% and 46.5%, the MAE was reduced by 19.9% and 54.5%, and the correlation coefficient R was improved by 9.88% and 29.5%, respectively. The results showed that the prediction performance of all three model architectures was significantly improved by adding the dummy variable, which represents spatial heterogeneity. After adding the dummy variable, RMSE decreased by 27.5% and 20.4%, MAE decreased by 18.4% and 20.0%, and the correlation coefficient R improved by 7.6% and 16.1%, respectively. The inclusion of dummy variable is very effective in predicting yield.

Deep learning model architectures are widely used in the field of crop yield prediction. Among the most frequently applied models are CNN models and LSTM models. Many studies have combined CNN and LSTM to improve the accuracy of crop yield prediction. Positive feedback has already been received. Sun proposed a hybrid CNN-LSTM model architecture to predict U.S. county-level soybean yields using MODIS surface reflectance and surface temperature, as well as weather data. The CNN-LSTM hybrid model was superior in yield prediction for each year from 2011 to 2015. The average RMSE of the CNN-LSTM hybrid model was reduced by about 8.24% and 9.26%, respectively, compared to the single CNN and LSTM [31]. Ghazaryan used the MODIS time series of surface reflectance, surface temperature and evapotranspiration as input datasets to test the prediction performance of three algorithms: random forest, 3D-CNN and CNN-LSTM. The results showed that for county-level analysis, the CNN-LSTM model had the highest accuracy. The average percentage error was 10.3% and 9.6% for corn and soybeans, respectively [48]. S Gastli predicted crop yields for the Midwestern states of the United States. CNN, LSTM, CNN-LSTM and CNN-LSTM integrated model architectures were tested using MODIS surface reflectance, surface temperature and humidity as input data. The results showed that the model architecture of two CNN-LSTM ensembles performed best in predicting annual soybean yields. It improved RMSE by 31% [49]. Xiang used NDVI, EVI, surface temperature and soil moisture as feature variables to train and predict corn yields in the United States from 2001 to 2018 based on a CNN-LSTM hybrid model. The results showed that the hybrid model had the smallest prediction error [24].

In the comparison experiments, we found that the ConvLSTM model had the worst prediction effect, even inferior to the prediction effect of a single CNN model. Yaramasu used a CNN model to extract spatial features before using ConvLSTM, and the extracted features were passed to ConvLSTM to extract temporal features [30]. Nevavuori also obtained the worst results for ConvLSTM prediction when testing the prediction performance of ConvLSTM, 3D-CNN and CNN-LSTM model architectures [26]. He thought, based on

Yaramasu's experiments, that it was likely that ConvLSTM would also require a pretrained CNN similar to the CNN-LSTM hybrid model. On the other hand, the CNN-LSTM model is more complex compared to the ConvLSTM model and is more advantageous for training a large number of datasets.

We conducted comparison experiments of three models and found that the models based on CNN have better prediction performance in the rice yield prediction problem. Compared with LSTM, CNN utilizes convolution kernel for convolution operation. Preemptive spatial features before processing the data of time series can produce more accurate yield prediction results. However, the single CNN model is not as good as the model combining CNN and LSTM in terms of the final prediction results. This is because the data input to our experiments is a long time series. Moreover, the time interval chosen is the growing period of double-season rice, and the input data have some correlation in time. A single CNN does not handle the data of long time series well. Due to the existence of the feedback structure of LSTM and the use of the gate function mechanism, it deals well with the problem of data gradient explosion. It is more effective in dealing with large number of time series. In our experiments, CNN and LSTM are combined to predict rice yield. It can be found that the hybrid CNN-LSTM model has better prediction performance. The hybrid model uses CNN to pre-extract the spatial features of the input variables and adds LSTM to make up for the deficiency of CNN, which finds it difficult to handle long time series data. Compared with CNN, ConvLSTM also incorporates convolutional structure, but the convolutional effect is not as good as CNN, so the performance in the accuracy of yield prediction is also not as good as CNN. In addition, ConvLSTM has a more homogeneous structure compared with the CNN-LSTM hybrid model. There is a disadvantage when dealing with a large amount of rice feature data. The CNN-LSTM model performs better when processing and predicting data with a large dataset of long time series. Our experimental results also argued this point.

In the field of crop yield prediction, the effect of spatial heterogeneity has been less considered in existing studies. Spatial heterogeneity is a field of study in ecology. In the field of ecology, many studies have shown that spatial heterogeneity has an effect on ecological environmental variables and soil variables, among others. Since ecology and soil are very important for crop growth, these effects can indirectly affect crop yield. In his study on the influence of spatial heterogeneity on ecosystem resilience in the Yangtze River basin, Liu Xiaofu found that heterogeneities such as rainfall, topography (elevation, slope) and human activities all have different degrees of influence with ecosystems [50]. China has diverse climate types, and farmland field management practices are diverse and complex in all regions. This resulted in a large spatial variation in soil respiration in agricultural fields at the regional scale. Han Guangxuan studied this phenomenon. It was found that field management such as tillage, fertilization, drainage and irrigation can directly or indirectly influence the spatial and temporal distribution of crop growth and environmental factors, thus affecting the spatial heterogeneity of soil respiration [51]. Ma proposed the concept of dynamic STR-NVI space based on high spatial and temporal resolution Sentinel-2 images to quantify the spatial heterogeneity of soil moisture and vegetation conditions. The results showed that dynamic STR-NVI space can improve the interpretation of soil moisture spatial heterogeneity at the farm scale and support irrigation and crop growth monitoring efforts in precision agriculture [52].

## 5. Conclusions

Timely and accurate yield forecasting is important for ensuring food security and making reasonable crop planting plans. This paper tested three deep learning model architectures based on a deep learning approach to predict rice yields in Hubei Province, China. In our study, we also considered the effect of spatial heterogeneity within regions on rice yield. The performance of a custom dummy variable in rice yield prediction was evaluated. Through the experiments, we justified the research objectives and obtained the following conclusions. (1) The hybrid CNN-LSTM model performed more satisfactorily

in rice yield prediction compared to the CNN and ConvLSTM models. The CNN-LSTM hybrid model had the best results on all three evaluation metrics with root mean square error RMSE = 89,878 (t), mean absolute error MAE =5 2,802 (t) and correlation coefficient R = 0.934. (2) Spatial heterogeneity has an impact on yield prediction. The self-defined dummy variables representing county numbers can improve the accuracy of rice yield prediction results.

In considering the effect of spatial heterogeneity on crop yield, customizing the dummy variable for county numbering was only a preliminary attempt at this idea. Some improvement methods may be considered in future work by refining the effect of spatial heterogeneity on crop yield and proposing more complex features or models for representing spatial heterogeneity.

There are also aspects of this study that could be improved in terms of yield prediction. First, only EVI, GPP, SAVI and air temperature data are used for training and prediction, but climate-related features are lacking. In the future, factors such as soil moisture, sunshine, precipitation and other factors related to rice growth can be added to the input dataset. Second, the remote sensing images are converted into normalized histograms, and the relevant spatial feature information may be lost in the histogram conversion process. In addition, in this study, we set the number of bins in the histogram to 32, and the number of bins has an effect on the training of the dataset. Future work can consider methods for combining multiple sources of data with different resolutions and time steps. Comparative experiments on the prediction effectiveness of different bin numbers can be conducted to explore the best performing histogram bin numbers. Third, modeling based on purely remote sensing data does not take into account existing crop growth models. The integration of remote sensing data with crop growth models can be considered, which may further enhance the predictive performance of the models.

**Author Contributions:** Conceptualization, S.Z., L.X. and N.C.; methodology, S.Z. and L.X.; software, S.Z.; validation, S.Z. and L.X.; formal analysis, S.Z.; data curation, S.Z.; writing—original draft preparation, S.Z.; writing—review and editing, S.Z., L.X. and N.C. All authors have read and agreed to the published version of the manuscript.

**Funding:** This project was supported by the Fundamental Research Funds for the Central Universities, China University of Geosciences (Wuhan) (No. 2642022009), the Special Fund of Hubei Luojia Laboratory (No. 220100034), and the National Natural Science Foundation of China (No. 42201509).

**Data Availability Statement:** Not applicable.

**Conflicts of Interest:** The authors declare no conflict of interest.

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
