# Peer review of "Rice Yield Prediction in Hubei Province Based on Deep Learning and the Effect of Spatial Heterogeneity"

_remotesensing, doi:10.3390/rs15051361_

Round 1

Reviewer 1 Report

1. Study objectives  should be clearly define and answered in the results and the conclusion

2. Rice crop model was used in the study should be clearly descripted and evaluate the reason for its application instead of other models

3. The linking of rice crop simulation results and depth learning  and remote sensing should be interpreted with some advantage and disadvantage evaluation as the accuracy with significance, instead of results description only  

 4. The results from linking of remote sensing results interpreted from image processing (Figure 1) and crop model  should be delineated on maps, instead of Figure 4 with GIS maps only 

5. The process for rice planting area mapping (Figure 1) should be descripted with accuracy assessment results 

Reviewer 2 Report

In General

This is nice paper about rice yield prediction which is very important for food security and food trade especially plausible estimates of climate change contributions to future yield growth have become most important. And it is written very well. However, the paper need to check carefully for the spelling, grammar and unit used such as:

L46: “lansat” should be Landsat

L47: “Soil-“ should be Soil

And some questions need to clear.

In details:

Figure 4: what is the unit of LOSS and MAE. And so to Figure 7 and L468-469.

Reviewer 3 Report

This paper is a good case study for predicting rice yield in Hubei Province based on deep learning methods. However, some questions have not been well addressed. Thus, my suggestion is major revision at this stage.

1. Line 17, what is the ConvLSTM, please clarify it. 

2. Line 46, lansat or Landsat? 

3. Line 64-104, Could you give more examples of yield prediction for comparing the machine learning and deep learning methods? because machine learning methods are widely used in this area. It's hard to say which of these two methods is better.

4. Line 165, Have you considered the fluctuation of rice area from 2000 to 2019?

5. Line 200, what is the spatial resolution of ERA5 for this paper, because it has different versions. why not use station-based climate data?

6. Line 284, Some information is missing.

7. Discussions, I suggest that the author can give some more reasons to explain the advantages and disadvantages of each model, and the resulting differences in simulation results.

8. The type of the reference should be revised according the rule of journal.

9. Some grammatical errors and irregular English writing can be found in the full text. I suggested that authors seek a native speaker to improve them. 

Round 2

Reviewer 3 Report

Authors have revised the MS. I think it can be accepted at this stage.